# Assessing equity in the uptake of remote foot temperature monitoring in a large integrated US healthcare system

Alyson J. Littman[1,2,3]*, Andrew K. Timmons[1], Kenneth T. Jones[4], Suzanne Shirley[5], Jeffrey Robbins[6], Ernest Moy[4]

1 Seattle Epidemiologic Research and Information Center, Department of Veterans Affairs Puget Sound Health Care System, Seattle, WA, United States of America, 2 Seattle-Denver Center of Innovation for Veteran-Centered and Value-Driven Care, Health Services, Seattle, WA, United States of America, 3 Department of Epidemiology, University of Washington School of Public Health, Seattle, WA, United States of America, 4 VA Office of Health Equity, Washington, DC, United States of America, 5 VHA Innovation Ecosystem, Washington, DC, United States of America, 6 Specialty Care Services, Podiatry Program Office, VA Central Office, Washington, DC, United States of America

* Alyson.littman@va.gov

**Data Availability Statement:** The United States Department of Veterans Affairs (VA) places legal restrictions on access to veteran's health care data, which includes both identifying data and sensitive

## Abstract

### Objective

We assessed equity in the uptake of remote foot temperature monitoring (RTM) for amputation prevention throughout a large, integrated US healthcare system between 2019 and 2021, including comparisons across facilities and between patients enrolled and eligible patients not enrolled in RTM focusing on the Reach and Adoption dimensions of the Reach, Effectiveness, Adoption, Implementation, and Maintenance (RE-AIM) framework.

### Material and methods

To assess whether there was equitable use of RTM across facilities, we examined distributions of patient demographic, geographic, and facility characteristics across facility RTM use categories (e.g., no RTM use, and low, moderate, and high RTM use) among all eligible patients (n = 46,294). Second, to understand whether, among facilities using RTM, there was equitable enrollment of patients in RTM, we compared characteristics of patients enrolled in RTM (n = 1066) relative to a group of eligible patients not enrolled in RTM (n = 27,166) using logistic regression and including all covariates.

### Results

RTM use increased substantially from an average of 11 patients per month to over 40 patients per month between 2019 and 2021. High-use RTM facilities had higher complexity and a lower ratio of patients per podiatrist but did not have consistent evidence of better foot-care process measures. Among facilities offering RTM, enrollment varied by age, was inversely associated with Black race (vs. white), low income, living far from specialty care, and being in the highest quartiles of telehealth use prior to enrollment. Enrollment was

patient information. The analytic data sets used for this study are not permitted to leave the VA firewall without a Data Use Agreement. This limitation is consistent with other studies based on VA data. However, VA data are made freely available to researchers behind the VA firewall with an approved VA study protocol. For more information, please visit https://www.virec.research.va.gov or contact the VA Information Resource Center (VIReC) at vog.av@CeRIV.

**Funding:** This work was supported in part by the Department of Veterans Affairs (VA) Health Services Research and Development and the VA Office of Health Equity (CIN 13-402). The funders had no role in study design, data collection and analysis, decision to publish, or preparation of the manuscript.

**Competing interests:** The authors have declared that no competing interests exist.

positively associated with having osteomyelitis, Charcot foot, a partial foot amputation, BMI$\geq$30 kg/m$^2$, and high outpatient utilization.

## Conclusions

RTM growth was concentrated in a small number of higher-resourced facilities, with evidence of lower enrollment among those who were Black and lived farther from specialty care. Future studies are needed to identify and address barriers to uptake of new interventions like RTM to prevent exacerbating existing ulceration and amputation disparities.

## Introduction

Foot ulceration is a common complication of diabetes, a chronic condition that is a highly prevalent among US military Veterans [1]. Foot ulceration negatively impacts mobility and quality of life. With comprehensive care, only about 77% of ulcers heal within a year and recurrence is common [2]. Furthermore, individuals who are Black, Hispanic, and Native American [3–5] or live in rural areas [5, 6] have higher rates of ulceration and amputation than those who are white and live in urban areas, respectively. The reasons for the disparities are not well understood, but proposed reasons include more advanced presentation of foot problems and socioeconomic factors that may impact access to care [5–7]; systemic racism is likely a major contributing factor [8].

Elevated skin temperatures are an early sign of damage and in randomized controlled trials, risks of ulceration were *24% to 90% lower* in the foot temperature monitoring groups compared to the usual care control groups [9–13]. Monitoring of foot skin temperatures is now recommended in several clinical practice guidelines [14–16]. Despite guideline endorsement, foot temperature monitoring is rarely practiced because the approach tested in the trials, which involved a handheld thermometer and individual tracking and comparison of temperatures, is time consuming and onerous. Fortunately, new technologies, including temperature sensing mats, "smart" insoles/socks, and smartphone applications have made foot temperature monitoring easier [17, 18].

This study focuses on the daily-use telemedicine foot temperature monitoring SmartMat made by Podimetrics; Somerville, Massachusetts, USA. The in-home foot temperature monitoring requires no configuration or set up by the patient. A temperature scan takes 20 seconds, and the temperature data are transmitted to the cloud using an embedded cellular component; patients are not required to have Wi-Fi or home cellular service. The software detects "hot spots", defined as asymmetries of $\geq$2.2˚C between the same region on the left and right foot or different regions on the same foot. Temperature asymmetries that persist for at least two days are predictive of ulceration [19], including in those with foot deformities and partial foot amputations [20]. When a hot spot is detected, the company notifies and works with the patient to change behaviors (e.g., reduce standing/walking, wear protective footwear, and check feet for injury/infection). If the temperature asymmetries persist, the company notifies the patient's healthcare provider who determines the next steps, often including an in-person examination.

The Veterans Health Administration (VHA), the largest integrated healthcare system in the United States, began national implementation of remote foot temperature monitoring using Smartmats in 2019. Over 6 million Veterans received care from VHA in 2019 at one of over 170 medical centers and 1200 outpatient facilities [21]. The mean and median number of patients at each medical center (including satellite outpatient clinics) was 52,162 and 44,637 (interquartile range: 31,40 to 64,881), respectively [22]. The VHA Innovation Ecosystem [23]

launched the Initiative to End Diabetic Limb Loss [24] in partnership with the VHA Podiatry Service, Office of Health Equity and Office of Connected Care to design new care models that incorporated emerging technologies like the SmartMat in early detection of diabetic foot ulcers. This initiative encouraged providers (primarily podiatrists) to enroll patients in remote foot temperature monitoring, a program that was offered free of charge to the Veteran. The standard operating procedures in place in 2019 detailed circumstances for appropriate provision of temperature monitoring devices (Kyle Nordrum, personal communication). These guidelines declared suitability for RTM was based on patients having either 1) peripheral neuropathy with peripheral artery disease and/or a foot deformity or 2) peripheral neuropathy and history of foot ulcer or lower extremity amputation.

In a separate report, we assessed effectiveness of RTM in VA; compared to usual care, RTM was not associated with a reduction in lower extremity amputation or hospitalization, though it was associated with a reduced risk of death [25]. Although that study [25] was unable to determine the reasons for the absence of a reduction in amputation risk, hypothesized reasons include patients not using the SmartMat as directed, patients not having been alerted when there was a hot spot or not complying with instructions when alerted, and/or enrolling patients who may not benefit from RTM (e.g., patients with peripheral artery disease who would not mount an inflammatory response to injury or infection). A recent study of patients who monitored foot skin temperatures via a handheld thermometer found that skin temperatures were frequently not elevated prior to ulcer development, calling into question the foot temperature increase-ulcer association [26]. Given these confusing results, more research is needed to confirm or refute the recent findings and to help illuminate the reasons for the discrepancies; a multi-site randomized controlled trial is underway to rigorously evaluate outcomes (and mediators of outcomes) [27].

Meanwhile, because RTM has potential to reduce the risk of ulceration and amputation, assessing the extent to which it has been implemented in proportion to need is valuable. Thus, the goal of this study was to evaluate reach and adoption of remote temperature monitoring (RTM) in VHA, using the Reach, Effectiveness, Adoption, Implementation, and Maintenance (RE-AIM) framework [28]. The RE-AIM framework was designed to make research findings more generalizable by encouraging scientists and evaluators to balance internal and external validity when developing and testing interventions. Reach is typically defined as the number, proportion of the intended audience, and the representativeness of participants compared with the intended audience. Adoption is defined as the number and proportion of settings and staff members that agree to initiate program or policy change and how representative they are of the intended audience. We evaluated adoption by first describing the geographic distribution of RTM throughout VHA over time and then assessing difference in characteristics of patients who received care at facilities with higher vs. lower (or no) use of RTM. The goal of these adoption analyses was to assess whether there were differences in facility-level characteristics based on RTM use. Next, to understand whether there was equitable use of RTM in the facilities that employed RTM, we compared patient characteristics of those enrolled in RTM relative to a comparison group of patients eligible for, but not enrolled in RTM (reach analyses). Information from this study can be used to identify geographic areas and subpopulations that may be underutilizing this technology and may benefit from interventions to increase uptake.

## Materials and methods

### Study design, population, and data source

We conducted an observational study using national data from VHA electronic medical records. Data were accessed between July 2020 and December 2022. To be eligible for inclusion,

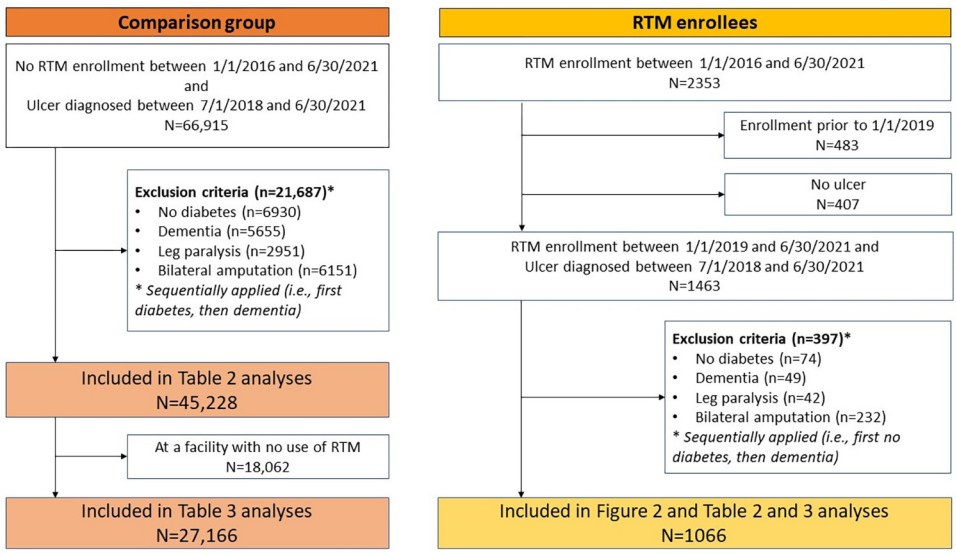

**Fig 1. Diagram describing inclusion and exclusion criteria for study analyses.** The figure shows the number of people included in the Table 1 and Table 2 analyses. Table 2 includes 46,294 patients, including 45,228 who never enrolled in RTM and 1066 who were enrolled in RTM.

patients had to have diabetes and at least two diagnosis or procedure codes for an ulceration between July 1, 2018, and June 30, 2021 (Fig 1). We chose July 1, 2018 to allow for at least six months before the first possible enrollment date for these analyses. These criteria applied both to patients enrolled and not enrolled in RTM. Current procedural terminology and International Classification of Diseases (ICD)-9 and ICD-10 codes were used to identify ulcers. See S1 Table for specific codes. The date of the earliest encounter for the ulcer was used as the baseline date for assessing covariates unless noted otherwise. We excluded patients from both groups if they had codes in the two years prior to baseline indicating that they were not good candidates for RTM including: dementia; inability to walk, as determined by diagnosis codes for quadriplegia, paraplegia, hemiplegia, and/or spinal cord injury; or bilateral major lower extremity amputation (because they did not have at least one foot that could be monitored via temperature scans). We also excluded patients if they had no primary care encounters in the two years prior to baseline, or they were enrolled in RTM prior to January 1, 2019.

## RTM enrollees

We identified patients who enrolled in RTM between January 1, 2019, and June 30, 2021 based on the unique vendor Data Universal Number System & Bradstreet ("DUNS") number for Podimetrics. We selected January 1, 2019 as the starting date because it corresponds to a point in time when RTM was being used at numerous medical centers throughout VHA. To examine adoption, we classified facilities based on the proportion of eligible patients enrolled in RTM during the study time period into one of four categories: no RTM use, low RTM use (<2% of eligible patients enrolled), moderate RTM use (2-<10% of eligible patients enrolled), and high RTM use (≥10% of eligible patients enrolled).

## Covariates

We evaluated demographic, geographic, clinical, and facility factors, as well as utilization. Details about the data sources, definitions, and categories are included in Table 1.

**Table 1. Covariates.**

| Domain, variable | Categories | Data source; other details |
|---|---|---|
| *Demographics* | | *CDW* |
| Sex | Male, female | Assumed to be sex assigned at birth |
| Race | American Indian/Alaska Native, Asian, Black/African American, Native Hawaiian/Pacific Islander, White, more than one race, unknown | |
| Hispanic/Latinx ethnicity | Yes, no, unknown | |
| Priority group | Co-payments required (Priority groups 7 and 8), low income (Priority group 5), moderate disability (Priority groups 2, 3, and 6), severe disability (Priority groups 1 and 4). | Veterans assigned to one of eight priority groups at the time of enrollment, determined based on military service history, disability rating, income level, Medicaid eligibility, and other criteria. We used VA priority group as a proxy for low income and disability status using the approach developed by other researchers [29] |
| *Geographic access* | | *CDW; Drive distance and drive time used the most recent data (FY14-FY19)* |
| Drive distance to primary care | <40 miles, ≥40 miles | Network distance between the coordinates of the primary care facility nearest to the Veteran's home and the Veteran's home address. |
| Drive time to primary care | <30 minutes, ≥30 minutes | Estimated drive time between the coordinates of the primary care facility nearest to the Veteran's home and the Veteran's home address. |
| Drive time to specialty care | <60 minutes, ≥60 minutes | Estimated drive time (network distance) between the coordinates of the specialty care facility nearest to the Veteran's home and the Veteran's home address. |
| Rurality | Urban, rural, highly rural | Determined using Rural Urban Commuting Area (RUCA) codes, which are based on zip code approximations. RUCA codes of 10.0 = highly rural; RUCA codes of 1.0 or 1.1 = urban. All other codes were considered rural [30]. |
| VA districts | Continental, Midwest, North Atlantic, Pacific, and Southeast | |
| *Area level factors* | | |
| Area deprivation index (ADI) | Quartiles–lower ADI indicates *less* deprivation | The ADI is a measure of socioeconomic resources and well-being that includes factors for income, education, employment, and housing quality. The ADI has been adapted and validated to the Census Block Group [31] and allows for rankings of neighborhoods by socioeconomic disadvantage at the national level. |
| Fixed broadband access (3G) | <25% no broadband ("good" access), ≥25% no broadband ("poor" access) | 2019 Federal Communications Commission data; Measured at the Census Block as percent of square miles in the census block with fixed broadband access |
| *Clinical characteristics* | | *CDW; Ascertained in the 2 years prior to qualifying condition (except for qualifying condition). See Supplemental File 1 for ICD/CPT codes* |
| Foot conditions: Ulcer, osteomyelitis, Charcot foot, lower extremity amputation | Yes, no | At least two diagnosis codes or one procedure code between July 1, 2018, and June 30, 2021. |
| Other conditions: Chronic kidney disease/end stage renal disease, diabetes, and depression | Yes, no | At least two diagnosis codes or one procedure code. |
| Gagne comorbidity index | <0, 1–2, 3–4, >4 | Measure of comorbidity burden. Higher scores indicate more comorbidities [32]. |
| Body mass index | <18.5, 18.5–24.9, 25.0–29.9, 30.0–39.9, ≥40 kg/m$^2$ | Weight and height measured closest to and prior to the qualifying condition date |
| *Facility characteristics* | | *CDW* |

(*Continued*)

**Table 1.** (Continued)

| Domain, variable | Categories | Data source; other details |
| --- | --- | --- |
| Above the knee (AK)/ below the knee (BK) amputation | Quartiles at the facility level | AK/BK ratio is the ratio of AK amputation daily procedures to BK amputation daily procedures minus 1. Used as a proxy measure of care quality; scores <0 reflect desirable results (proportionally fewer AK amputations) and scores >0 may signal an undesirable trend in limb salvage success efforts. |
| Percentage of patients at a facility whose sensation was evaluated via a monofilament exam | Quartiles at the facility level | Assessment of protective sensation using the Semmes-Weinstein 5.07 monofilament is recommended on an annual basis for all patients with diabetes [33]. |
| Facility complexity | 1a (most complex), 1b, 1c, 2, 3 (low complexity) | Determined based on a model that considers clinical programs and patient risk levels, as well as research and teaching. The model is reviewed and updated with current data every 3 years. |
| Percent of patients classified as at high risk of amputation | Quartiles (at the facility) of % high risk | High-risk patients (Preventing Amputation in Veterans Everywhere [PAVE] risk = 3). PAVE 3 patients have ≥1 of the following: history of prior ulcer, osteomyelitis, or prior amputation; severe peripheral arterial disease (e.g., critical limb threatening ischemia); Charcot's joint disease with foot deformity; chronic kidney disease. |
| Number of patients per podiatrist | Quartiles at the facility level | Size of the podiatry department at each VHA facility, determined based on FY19 data, divided by the total number of unique patients at that facility in FY19 |
| *Utilization* | | *CDW; Ascertained in the two years prior to the qualifying condition date.* |
| Podiatry, wound care, or physical therapy encounter | Percentage with at least one encounter | Stop codes (internal billing codes) used to identify podiatry (411), physical therapy (20, 205), and wound treatment and care (142, 301) |
| Outpatient encounters | Quartiles | |
| Inpatient stays | 0,1+ | |
| ER/urgent care | 0,1, 2+ | Stop codes 130 and 131 |
| Telehealth encounters | Quartiles | Determined by stop codes, and includes visits that were designated as telephone, video, "tele", and/or virtual |

CDW, Corporate Data Warehouse

## Statistical analyses

To understand the geographic distribution of RTM throughout VHA over time, we mapped the number of RTM enrollees based on each patient's facility location. To assess whether there was equitable use of RTM across facilities, we examined distributions of patient demographic, geographic, and facility characteristics across facility RTM use categories (e.g., no RTM use, and low, moderate, and high RTM use) and calculated chi-square statistics to test the statistical significance of differences. Because statistical tests with so many comparisons and a large sample may not be informative in terms of meaningful differences, we highlighted meaningful differences between categories (see Results). These analyses include all patients classified as eligible at these facilities, not just those enrolled in RTM.

To understand whether, among facilities using RTM, there was equitable enrollment of patients in RTM, we compared characteristics of patients enrolled in RTM relative to a group of eligible patients not enrolled in RTM. We estimated odds ratios and corresponding 95% confidence intervals of enrollment using a logistic regression model that included all covariates. Akaike information Criterion (AIC) was used to assess contributions of each covariate and group of covariates to model fit using a likelihood ratio test for a model that excluded the covariate or group of covariates. AIC is a statistic for evaluating how well a model fits the data it was generated from relative to other models fit on the same data. AIC penalizes models that

use more parameters to reduce the potential for overfitting, and lower AIC scores are considered evidence of better model fit. Additionally, to evaluate whether selection criteria may differ between facilities, we conducted an exploratory sensitivity analysis stratifying bivariate RTM enrollment and patient characteristics on facility RTM use (low, medium, high). As the relationships were relatively consistent across strata and because of the additional complexity of the stratified results, we present only the unstratified results.

Missing data were accounted for using multiple imputation by chained equations using all covariates and the outcome and pooling results from 20 imputed data sets [34–36]. Generalized variance inflation factors [37] for each of the covariates were calculated to assess correlation between covariates, and the impact it may have on regression results. A variance inflation factor of 4 or more was used as evidence of substantial collinearity [38].

### Ethics approval

This program evaluation qualified as non-research quality improvement activity conducted under the authority of Veterans Health Administration (VHA) operations. Consequently, Institutional Review Board review or approval was not sought and the need for patient consent was waived. This program evaluation complies with the VHA definition of "non-research operations activities" outlined in section 5a of the 2019 *VHA Program Guide 1200.21*: *VHA Operations Activities That May Constitute Research*, meeting both specified conditions: (1) the evaluation was designed and implemented for internal VHA purposes and (2) not designed to produce information to expand the knowledge base of a scientific discipline.

### Results

We identified 1066 patients who met study inclusion criteria and were enrolled in RTM and 45,228 patients who met study inclusion criteria but were not enrolled in RTM. Of the 45,228, 27,166 patients were at a facility where at least one patient was enrolled in RTM (Fig 1). Fig 2 shows how RTM use varied over time and was distributed across VHA facilities. Each map

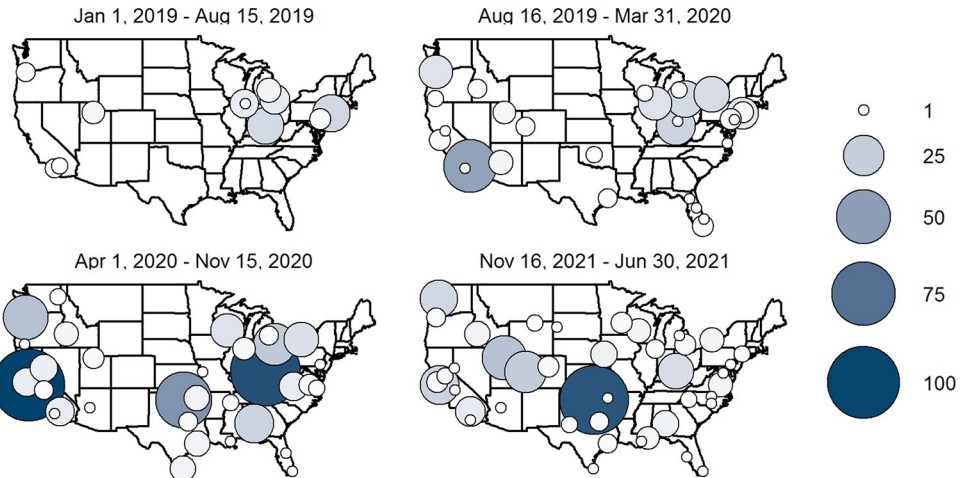

**Fig 2. Remote foot temperature monitoring enrollment over time by facility.** The map on the top left represents enrollments between January 1, 2019, and August 15, 2019. The map on the top right represents enrollment between August 16, 2019, and March 31, 2020. The map on the bottom left represents enrollment between April 1, 2020, and November 15, 2020. The map on the bottom right represents enrollment between November 16, 2020, and June 30, 2021. "Parent" facilities may include more than one medical center. For example, the Palo Alto VHA Medical Center parent facility includes the Palo Alto VHA, as well as facilities in Menlo Park and Livermore and the Portland VHA Medical Center parent facility includes facilities in Portland, OR as well as Vancouver, WA.

covers a 7.5-month period. Average monthly enrollment per facility among patients who met this evaluation's eligibility criteria was 11.2 during the first 7.5-month period (January 1, 2019-August 15, 2019); 26.9 patients per month in the second time period; 61.7 patients per month in the third time period, which began shortly after the COVID-19 pandemic was declared; and 42.3 patients per month in the fourth time period. The number of facilities that enrolled at least 10 patients (1.3 per month) over a 7.5-month time period was 4, 7, 11, and 7, during the first, second, third, and fourth time periods, respectively, including 8 facilities that enrolled at least 10 patients in more than one time period. Notably, in the third time period (April 1, 2020-November 15, 2020), the top two facilities (Cincinnati and Palo Alto VHA Medical Centers) each enrolled more than 12 patients per month (13.5 patients per month in Palo Alto and 12.4 in Cincinnati), while the next three highest facilities enrolled 7.3, 4.3, and 3.6 patients per month, respectively. In addition to an increase in the number of facilities that enrolled 10 patients or more during a time period, the number of facilities that enrolled 1–9 patients increased over time, from 7 facilities in the first time period to 35 in the fourth time period. Fig 2 also makes clear that there are several states with no patients enrolled in RTM (e.g., Montana, North and South Dakota, Maine, New Hampshire, and Vermont). Additionally, facility growth in RTM was not linear or predictable, as some facilities, like the Manhattan VHA, enrolled 2.7 patients per month in the first time period, 1.7 patients per month in the second, and none in the third and fourth time periods.

Among the 46,294 patients eligible for RTM, 39%, 36%, 15% and 10% received care from facilities with no, low, moderate, and high RTM use, respectively (Table 2). Differences across facility categories were statistically significant for all variables. Here, we highlight larger absolute differences across categories (≥5 percentage points between high and no patient categories). Although there was variation across use groups in terms of race and ethnicity, the differences were relatively small and did not show a clear trend. Compared to patients at non-high-use facilities, patients at high-use facilities had better geographic access to care based on drive-time. Patients at high-use facilities were disproportionately urban and in the Pacific region, and small proportion of patients at moderate- and high-use facilities were in the Southeast. Based on the Area Deprivation Index, a composite measure of deprivation where lower scores represent *less* deprivation, a greater proportion of patients at high-use facilities lived in areas with low deprivation.

Lastly, a greater proportion of the high-use facilities were high complexity and had lower patient to podiatrist ratios (40.6% in the lowest quartile in high-use facilities vs. 23.7% in the lowest quartile for patients at facilities with no patients enrolled). Findings were mixed in terms of quality measures. Specifically, high-use facilities had better above the knee (AK) to below the knee (BK) amputation ratios (when considering the lower two quartiles), possibly indicating better care quality, though patients at high-use facilities were *less* likely to have been at the best performing facilities based on the proportion of patients who were assessed for loss of protective sensation (a second measure of quality).

Next, we compared patient characteristics among those who were enrolled and not enrolled in RTM when only considering patients receiving care at facilities with at least one patient enrolled in RTM (Table 3). There was no evidence of substantial collinearity. Among eligible patients at facilities where at least one patient was enrolled in RTM, 3.7% (1006/27,166) were enrolled in RTM. In the multivariable model, factors that were independently and inversely associated (based on a 95% CI excluding 1.00) with RTM enrollment were age 80+ (vs. 70–79), Black race (vs. white), low income (vs. copayments required), living 60+ minutes' drive time from specialty care (vs. <60), and having a moderate to high number of telehealth encounters in the prior two years (2nd, 3rd, and 4th quartiles vs. the 1st). Factors that were positively associated with RTM enrollment were having osteomyelitis, Charcot foot, and a partial foot

**Table 2. Patient characteristics by site remote temperature monitoring patient volume (between January 1, 2019 and June 30, 2021) for all eligible patients.**

| | All eligible patients (n = 46,294) | | No mats (n = 18,062, 39%) | | Low (<2%) (n = 16,706, 36%) | | Moderate (2-<10%) (n = 6832, 15%) | | High (≥10%) (n = 4694, 10%) | |
|---|---|---|---|---|---|---|---|---|---|---|
| **Demographic characteristics** | | | | | | | | | | |
| **Race** | | | | | | | | | | |
| American Indian or Alaska Native | 1.0% | (439) | 0.8% | (146) | 0.9% | (149) | 1.5% | (94) | 1.1% | (50) |
| Asian | 0.3% | (113) | 0.2% | (28) | 0.3% | (43) | 0.5% | (30) | 0.3% | (12) |
| Black or African American | 17.1% | (7,557) | 15.9% | (2,742) | 18.3% | (2,918) | 18.1% | (1,169) | 16.4% | (728) |
| More than one race | 0.8% | (371) | 0.7% | (115) | 1.0% | (164) | 0.9% | (61) | 0.7% | (31) |
| Native Hawaiian or Other Pacific Islander | 0.8% | (345) | 0.7% | (125) | 0.8% | (129) | 0.8% | (51) | 0.9% | (40) |
| White | 80.0% | (35,303) | 81.7% | (14,136) | 78.7% | (12,542) | 78.3% | (5,057) | 80.6% | (3,568) |
| *Unknown* | *4.7%* | *(2,166)* | *4.3%* | *(770)* | *4.6%* | *(761)* | *5.4%* | *(370)* | *5.6%* | *(265)* |
| **Hispanic or Latinx ethnicity** | 5.7% | (2,622) | 6.2% | (1,127) | 4.2% | (698) | 7.4% | (508) | 6.2% | (289) |
| **Geographic access** | | | | | | | | | | |
| 40+ miles drive distance to primary care | 6.7% | (3,083) | 7.3% | (1,316) | 6.4% | (1,069) | 5.9% | (406) | 6.2% | (292) |
| 30+ minutes' drive-time to primary care | 20.6% | (9,532) | 23.7% | (4,287) | 18.9% | (3,154) | 20.0% | (1,368) | 15.4% | (723) |
| 60+ minutes' drive-time to specialty care | 25.6% | (11,869) | 29.5% | (5,332) | 24.2% | (4,035) | 23.2% | (1,583) | 19.6% | (919) |
| **Rurality** | | | | | | | | | | |
| Highly rural | 1.3% | (587) | 1.5% | (268) | 1.2% | (202) | 1.2% | (82) | 0.7% | (35) |
| Rural | 33.3% | (15,415) | 38.0% | (6,858) | 30.9% | (5,167) | 32.3% | (2,208) | 25.2% | (1,182) |
| Urban | 65.4% | (30,257) | 60.4% | (10,916) | 67.8% | (11,327) | 66.5% | (4,541) | 74.0% | (3,473) |
| **District** | | | | | | | | | | |
| Continental | 16.7% | (7,736) | 12.4% | (2,236) | 18.1% | (3,031) | 20.8% | (1,422) | 22.3% | (1047) |
| Midwest | 23.4% | (10,829) | 23.7% | (4,274) | 29.8% | (4,978) | 3.4% | (230) | 28.7% | (1,347) |
| North Atlantic | 22.0% | (10,177) | 28.0% | (5,066) | 14.4% | (2,410) | 35.7% | (2,439) | 5.6% | (262) |
| Pacific | 18.4% | (8,508) | 9.5% | (1,713) | 15.3% | (2,551) | 36.1% | (2,468) | 37.8% | (1,776) |
| Southeast | 19.5% | (9,044) | 26.4% | (4,773) | 22.4% | (3,736) | 4.0% | (273) | 5.6% | (262) |
| **Area deprivation index (national rank; lower value indicates *less deprivation*)** | | | | | | | | | | |
| 1–24 | 11.3% | (5,149) | 7.4% | (1,313) | 9.5% | (1,573) | 19.5% | (1,311) | 20.5% | (952) |
| 25–49 | 26.7% | (12,195) | 22.7% | (4,034) | 28.2% | (4,654) | 28.6% | (1,923) | 34.2% | (1,584) |
| 50–74 | 32.1% | (14,670) | 34.3% | (6,106) | 32.8% | (5,426) | 28.9% | (1,947) | 25.7% | (1,191) |
| 75+ | 29.9% | (13,671) | 35.6% | (6,341) | 29.5% | (4,870) | 23.1% | (1,552) | 19.6% | (908) |
| **Poor broadband access** | 16.6% | (7,701) | 17.3% | (3,120) | 16.1% | (2,688) | 17.5% | (1,196) | 14.8% | (697) |
| **Facility characteristics** | | | | | | | | | | |
| **Facility complexity** | | | | | | | | | | |
| 1a-High Complexity | 44.5% | (20,612) | 39.7% | (7,172) | 45.5% | (7,602) | 47.1% | (3,220) | 55.8% | (2,618) |
| 1b-High Complexity | 21.3% | (9,858) | 18.7% | (3,370) | 26.0% | (4,337) | 13.6% | (929) | 26.0% | (1,222) |
| 1c-High Complexity | 14.7% | (6,822) | 10.4% | (1,880) | 22.5% | (3,760) | 14.0% | (957) | 4.8% | (225) |
| 2-Medium Complexity | 10.0% | (4,612) | 15.9% | (2,866) | 1.9% | (325) | 17.0% | (1,159) | 5.6% | (262) |
| 3-Low Complexity | 9.2% | (4,255) | 14.6% | (2,639) | 4.1% | (682) | 8.3% | (567) | 7.8% | (367) |
| **Patients per podiatrist (quartiles)** | | | | | | | | | | |
| Q1: 2793 to 8013 | 22.2% | (10,288) | 23.7% | (4,279) | 18.5% | (3,086) | 14.9% | (1,018) | 40.6% | (1,905) |
| Q2: >8013 to 11907 | 26.2% | (12,139) | 29.8% | (5,379) | 18.9% | (3,156) | 34.0% | (2,322) | 27.3% | (1,282) |
| Q3: >11907 to 17090 | 27.4% | (12,668) | 22.5% | (4,059) | 41.3% | (6,906) | 21.6% | (1,478) | 4.8% | (225) |
| Q4: >17090 to 63177 | 23.2% | (10,763) | 22.1% | (3,992) | 20.8% | (3,475) | 29.5% | (1,478) | 27.3% | (1,282) |
| **AK/BK ratio (quartiles)** | | | | | | | | | | |
| Q1: -1.0 to -0.5 | 23.3% | (9,468) | 25.3% | (3,846) | 15.7% | (2,354) | 39.1% | (2,398) | 20.6% | (870) |

*(Continued)*

**Table 2.** (Continued)

| | All eligible patients (n = 46,294) | | No mats (n = 18,062, 39%) | | Low (<2%) (n = 16,706, 36%) | | Moderate (2-<10%) (n = 6832, 15%) | | High (≥10%) (n = 4694, 10%) | |
|---|---|---|---|---|---|---|---|---|---|---|
| Q2: >-0.5 to -0.4 | 28.8% | (11,687) | 29.5% | (4,496) | 26.0% | (3,908) | 20.9% | (1,283) | 47.5% | (2,000) |
| Q3: >-0.4 to -0.1 | 27.2% | (11,044) | 18.0% | (2,739) | 40.6% | (6,110) | 26.9% | (1,650) | 12.9% | (545) |
| Q4: >-0.1 to 2.1 | 20.7% | (8,412) | 27.2% | (4,144) | 17.7% | (2,666) | 13.1% | (803) | 19.0% | (799) |
| *Unknown* | *12.3%* | *(5,683)* | *15.7%* | *(2,837)* | *10.0%* | *(1,668)* | *10.2%* | *(698)* | *10.2%* | *(480)* |
| **% PAVE high risk (quartiles)** | | | | | | | | | | |
| Q1: 15% to 26% | 17.9% | (8,303) | 15.5% | (2,797) | 18.0% | (3,007) | 24.3% | (1,663) | 17.8% | (836) |
| Q2: >26% to 29% | 27.4% | (12,691) | 27.6% | (4,979) | 30.4% | (5,084) | 24.1% | (1,645) | 20.9% | (983) |
| Q3: >29% to 31% | 30.3% | (14,010) | 29.7% | (5,365) | 30.1% | (5,027) | 30.1% | (2,058) | 33.2% | (1,560) |
| Q4: >31% to 46% | 24.4% | (11,290) | 27.2% | (4,921) | 21.5% | (3,588) | 21.5% | (1,466) | 28.0% | (1,315) |
| **Percentage of patients with a documented monofilament exam (quartiles)** | | | | | | | | | | |
| Q1: 52% to 77% | 20.5% | (9,512) | 22.5% | (4,064) | 21.2% | (3,537) | 5.9% | (404) | 32.1% | (1,507) |
| Q2: >77% to 84% | 31.7% | (14,674) | 23.8% | (4,304) | 37.0% | (6,187) | 42.0% | (2,867) | 28.0% | (1,316) |
| Q3: >84% to 89% | 25.0% | (11,573) | 13.5% | (2,437) | 32.9% | (5,496) | 35.2% | (2,402) | 26.4% | (1,238) |
| Q4: >89% to 100% | 22.8% | (10,534) | 40.2% | (7,256) | 8.9% | (1,486) | 17.0% | (1,159) | 13.5% | (633) |

Numbers may not sum to totals because of missing data. Less than 1% missing for all measures unless otherwise noted; 3.1% missing for Hispanic/Latinx ethnicity; 1.3% missing for ADI. Percentages calculated among those with non-missing/non-unknown values.

All chi-square p<0.01 across categories.

amputation (vs. no amputation), intermediate Gagne comorbidity index (vs. the lowest category), body mass index ≥30 kg/m$^2$ (vs. 18.5–24.9), having more outpatient encounters (2nd, 3rd, and 4th quartiles vs. first) and 2 or more ER/urgent care visits (vs. none). Based on the AIC (Table 4), foot conditions (especially osteomyelitis and Charcot foot) were most strongly associated with RTM enrollment, with model fit deteriorating most when these covariates were removed. After foot conditions, utilization and demographics were the next conceptual categories most strongly associated with RTM enrollment, followed by other health conditions/comorbidities, and finally geographic access.

## Discussion

RTM use increased substantially over the 30-month observation period, including 15 months during the COVID-19 pandemic. Beginning in March 2020, VHA shifted from in-person encounters to virtual care (video and phone-based care) [39]. In fact, a study of Veterans 65 and older found that there were 824% more virtual visits in April-November 2020 compared to the pre-pandemic period [40]. RTM allowed providers to monitor patients' feet despite restrictions on in-person care; organizations like the International Working Group on the Diabetic Foot recommended temperature monitoring to more effectively care for patients with, or at risk for DFU when in-person care was limited [41]. Notably, though the number of patients enrolled and the number of facilities employing RTM increased substantially during the observation period, growth was concentrated in a small number of high-use facilities. RTM use was greater in facilities with a higher proportion of urban patients, lower area deprivation, higher complexity and those with a lower patient-to-podiatrist ratio. These findings indicate a need to identify and address barriers to RTM use in facilities with a high proportion of Veterans who live in rural areas or areas with high deprivation to prevent exacerbating existing disparities. Among facilities offering RTM, Black patients, those with a low income, and those with longer

**Table 3. Distribution and associations of demographic, geographic, and utilization characteristics among eligible VHA patients who were and were not enrolled in remote temperature monitoring between January 1, 2019, and June 30, 2021.**

| Characteristics | Enrolled in remote temperature monitoring | | | | Odds Ratio† | 95% CI |
|---|---|---|---|---|---|---|
| | No (n = 27,166) | | Yes (n = 1066) | | | |
| | % | N | % | N | | |
| **Age (years)** | | | | | | |
| <50 | 1.8% | (492) | 1.6% | (17) | 0.88 | 0.53, 1.45 |
| 50–59 | 10.4% | (2,821) | 11.8% | (126) | 1.06 | 0.86, 1.31 |
| 60–69 | 28.6% | (7,781) | 32.1% | (342) | 1.08 | 0.93, 1.26 |
| 70–79 | 44.7% | (12,153) | 46.7% | (498) | 1.00 | Ref |
| 80+ | 14.4% | (3,919) | 7.8% | (83) | **0.61** | **0.48, 0.78** |
| **Female** | 2.2% | (594) | 1.8% | (19) | 0.82 | 0.51, 1.30 |
| **Race** | | | | | | |
| Black or African American | 17.1% | (4,650) | 15.5% | (165) | **0.83** | **0.70, 0.99** |
| American Indian or Alaska    Native | 1.0% | (284) | 0.8% | (9) | 0.74‡ | 0.49, 1.12 |
| Asian | 0.3% | (85) | 0.0% | (0) | | |
| Native Hawaiian or other Pacific Islander | 0.9% | (248) | 0.8% | (8) | | |
| More than one race | 0.8% | (212) | 0.8% | (8) | | |
| White | 74.9% | (20,336) | 78.0% | (831) | 1.00 | Ref |
| *Unknown* | 5.0% | (1,351) | 4.2% | (45) | | |
| **Hispanic or Latinx** | 5.3% | (1,431) | 6.0% | (64) | 1.06 | 0.82, 1.39 |
| **Enrollment priority** | | | | | | |
| Copayments required | 12.5% | (3,391) | 13.5% | (144) | 1.00 | Ref |
| Low income | 25.4% | (6,909) | 20.8% | (222) | **0.74** | **0.60, 0.92** |
| Moderate disability | 16.0% | (4,334) | 17.0% | (181) | 0.96 | 0.77, 1.21 |
| Severe disability | 46.1% | (12,531) | 48.7% | (519) | 0.95 | 0.78, 1.16 |
| Foot conditions | | | | | | |
| **Osteomyelitis** | 21.9% | (11,461) | 37.7% | (463) | **1.67** | **1.46, 1.92** |
| **Charcot foot** | 8.4% | (2,289) | 15.1% | (161) | **1.61** | **1.35, 1.92** |
| **Lower extremity amputation** | | | | | | |
| None | 79.5% | (21,600) | 73.5% | (784) | 1.00 | Ref |
| Partial foot | 14.9% | (4,046) | 21.8% | (232) | **1.27** | **1.08, 1.49** |
| Major lower limb | 5.6% | (1,520) | 4.7% | (50) | 0.85 | 0.63, 1.15 |
| Other conditions | | | | | | |
| **CKD/ESRD** | 35.4% | (9,605) | 34.7% | (370) | 1.00 | 0.86, 1.16 |
| **Depression** | 26.5% | (7,208) | 26.5% | (282) | 0.92 | 0.79, 1.07 |
| **Hemoglobin A1c** | | | | | | |
| <5.7 | 4.9% | (1,329) | 5.0% | (53) | 1.00 | Ref |
| 5.7–6.9 | 28.1% | (7,637) | 25.9% | (276) | 0.92 | 0.68, 1.25 |
| 7.0–7.9 | 25.6% | (6,941) | 27.2% | (290) | 1.02 | 0.75, 1.38 |
| 8.0–9.9 | 27.3% | (7,421) | 29.8% | (318) | 0.97 | 0.72, 1.31 |
| 10.0+ | 10.7% | (2,913) | 10.6% | (113) | 0.86 | 0.62, 1.21 |
| *Unknown* | 3.4% | (925) | 1.5% | (16) | | |
| **Gagne index** | | | | | | |
| ≤0 | 33.0% | (8,971) | 28.0% | (298) | 1.00 | Ref |
| 1–2 | 16.3% | (4,419) | 20.0% | (213) | **1.23** | **1.02, 1.49** |
| 3–4 | 15.0% | (4,073) | 17.8% | (190) | 1.15 | 0.94, 1.41 |
| >4 | 35.7% | (9,703) | 34.2% | (365) | 0.97 | 0.79, 1.18 |
| **Body mass index (kg/m²)** | | | | | | |

*(Continued)*

**Table 3.** (Continued)

| Characteristics | Enrolled in remote temperature monitoring | | | | Odds Ratio[†] | 95% CI |
|---|---|---|---|---|---|---|
| | No (n = 27,166) | | Yes (n = 1066) | | | |
| | % | N | % | N | | |
| <18.5 | 0.6% | (165) | 0.3% | (3) | 0.72 | 0.22, 2.32 |
| 18.5–24.9 | 11.8% | (3,193) | 8.5% | (91) | 1.00 | Ref |
| 25.0–29.9 | 26.2% | (7,119) | 23.1% | (246) | 1.16 | 0.91, 1.49 |
| 30.0–39.9 | 44.1% | (11,968) | 52.3% | (557) | **1.50** | **1.19, 1.89** |
| 40.0+ | 12.5% | (3,385) | 13.0% | (139) | **1.33** | **1.01, 1.76** |
| *Unknown* | 4.9% | (1,336) | 2.8% | (30) | | |
| **Geographic access** | | | | | | |
| **40+ miles to primary care** | 6.3% | (1,704) | 5.9% | (63) | 1.18 | 0.86, 1.63 |
| **30+ minutes to primary care** | 18.6% | (5,063) | 17.1% | (182) | 0.98 | 0.79, 1.20 |
| **60+ minutes to specialty care** | 51.7% | (14,039) | 46.6% | (497) | **0.77** | **0.64, 0.92** |
| **Area with >25% no broadband (poor access)** | 16.3% | (4,428) | 14.4% | (153) | 0.91 | 0.76, 1.10 |
| **Utilization** | | | | | | |
| **Outpatient encounters** | | | | | | |
| <34 | 12.6% | (3,427) | 10.6% | (113) | 1.00 | Ref |
| 35–67 | 26.1% | (7,091) | 27.0% | (288) | **1.33** | **1.05, 1.69** |
| 68–115 | 30.6% | (8,322) | 30.3% | (323) | **1.43** | **1.10, 1.86** |
| 116+ | 30.6% | (8,326) | 32.1% | (342) | **1.74** | **1.29, 2.35** |
| **Telehealth encounters** | | | | | | |
| <6 | 17.6% | (4,773) | 20.9% | (223) | 1.00 | Ref |
| 6–12 | 22.1% | (6,003) | 22.4% | (239) | **0.73** | **0.60, 0.89** |
| 13–27 | 29.2% | (7,926) | 30.5% | (325) | **0.68** | **0.55, 0.84** |
| 28+ | 31.2% | (8,464) | 26.2% | (279) | **0.50** | **0.39, 0.64** |
| **Inpatient visits** | | | | | | |
| 0 | 62.8% | (17,054) | 61.3% | (653) | 1.00 | Ref |
| 1+ | 37.2% | (10,112) | 38.7% | (413) | 0.92 | 0.78, 1.08 |
| **ER/Urgent care visits** | | | | | | |
| 0 | 35.8% | (9,713) | 31.0% | (330) | 1.00 | Ref |
| 1 | 18.0% | (4,892) | 17.4% | (186) | 1.04 | 0.86, 1.26 |
| 2+ | 46.2% | (12,561) | 51.6% | (550) | **1.25** | **1.05, 1.49** |

Percentages were calculated among those with non-missing values.

[†] Estimated association of covariates with RTM enrollment using multiply imputed data (n = 28,232)

‡ Because of small numbers, American Indian/Alaska Native, Asian, Native Hawaiian or other Pacific Islander, and more than one race were combined into a single category.

Results that are statistically significant (p<0.05) are bolded.

drive times from specialty care were less likely to be enrolled in RTM. A greater proportion of patients enrolled in RTM had a second foot condition that put them at high risk of a future ulcer or amputation (osteomyelitis, Charcot foot, or a partial foot amputation). Those enrolled were no more likely to have other comorbidities such as CKD/ESRD, depression, or poor glucose control, but were more likely to have an intermediate level of comorbidities and obesity. Those enrolled were also more likely to be heavier utilizers of face-to-face healthcare services, suggesting that patients who are seen more frequently by clinicians are more likely to be invited and enrolled (opportunistic enrollment). If RTM lowers healthcare costs (a question we and others hope to answer in future research), targeting patients with high utilization may

**Table 4. Summary of Akaike information Criterion (AIC) for conceptual groups and variables.**

| Conceptual group | AIC | AIC difference vs. saturated model | F stat | p-value |
|---|---|---|---|---|
| **Saturated model** | 8873 | Ref | | |
| **Demographics** | 8892 | +19 | 3.7 | <0.001 |
| Age | 8888 | +15 | 5.7 | <0.001 |
| Sex | 8872 | -1 | 0.8 | 0.39 |
| Race | 8875 | +2 | 3.0 | 0.052 |
| Hispanic/Latinx ethnicity | 8871 | -2 | 0.2 | 0.65 |
| Enrollment priority | 8878 | +5 | 3.8 | 0.009 |
| **Foot conditions** | 8974 | +101 | 27.3 | <0.001 |
| Osteomyelitis | 8925 | +52 | 54.2 | <0.001 |
| Charcot foot | 8896 | +26 | 25.2 | <0.001 |
| Lower extremity amputation | 8880 | +7 | 5.4 | 0.005 |
| **Other health conditions/ comorbidities** | 8882 | +9 | 2.6 | 0.001 |
| CKD/ESRD | 8871 | -2 | 0.0 | 0.97 |
| Depression | 8872 | -1 | 1.1 | 0.29 |
| Hemoglobin A1c | 8868 | -5 | 0.7 | 0.62 |
| Gagne comorbidity index | 8876 | +3 | 2.9 | 0.036 |
| Body mass index | 8885 | +12 | 4.8 | 0.001 |
| **Geographic access** | 8876 | +3 | 2.9 | 0.021 |
| 40+ miles to primary care | 8872 | -1 | 1.0 | 0.31 |
| 30+ minutes to primary care | 8871 | -2 | 0.1 | 0.81 |
| 60+ minutes to specialty care | 8879 | +6 | 8.3 | 0.004 |
| Area with >25% no broadband (poor access) | 8872 | -1 | 0.9 | 0.34 |
| **Utilization** | 8895 | +22 | 4.5 | <0.001 |
| Outpatient encounters | 8880 | +7 | 4.5 | 0.004 |
| Telehealth encounters | 8898 | +25 | 10.4 | <0.001 |
| Inpatient visits | 8872 | -1 | 1.0 | 0.32 |
| ER/urgent care visits | 8876 | +3 | 3.7 | 0.024 |

Shading is used to indicate the variables or conceptual groups with larger positive changes to the AIC.

be a cost-effective strategy. However, to ensure equitable implementation of RTM, it will also be necessary to take a more systematic approach to enrolling patients.

To facilitate implementation of RTM during the period under study, the Innovation Ecosystem aimed to understand challenges providers and sites faced in incorporating RTM into preventative care (Suzanne Shirley, personal communication). The Innovation Ecosystem includes Innovator Network sites, which have designated "Innovations Specialists" who are clinicians, researchers, and administrators who work with frontline employees to identify, test, and spread innovative products and practices throughout VHA. The Innovation Ecosystem met with providers and learned that they found the ordering process to be complicated, time consuming and confusing. They also learned that there was insufficient knowledge about the availability and impact of RTM. The Innovation Ecosystem convened key stakeholders to identify potential solutions, which included approaches to make ordering easier; efforts to provide a stronger evidence base through evaluations of clinical, economic and equity related impacts (of which this study is a part); and clearer communication to providers on the impact and availability of this technology.

A few limitations should be considered when interpreting our results. First, identifying a comparison group was challenging. When evaluating appropriateness for RTM, clinicians

might consider not only whether a patient had a prior ulcer, but whether prior ulcers were on the plantar surface of the foot in addition to other factors. However, because ICD-10 codes do not specify ulcer location, we were not able to consider ulcer location as an inclusion criterion. Thus, we may have included patients who would not have been considered good candidates for RTM, including those with severe peripheral artery disease, as this condition is difficult to determine based on diagnosis and procedure codes and we lacked the resources to do chart review. Additionally, because we relied on diagnosis and procedure codes, we did not have data that might have been useful for risk adjustment, such as diabetes-related treatments, duration of diabetes, or time since last ulceration. Utilization measures were based solely on VHA care; we may have underestimated total healthcare utilization among patients who received some of their care outside VHA. Furthermore, codes may sometimes be applied erroneously. However, we have no reason to suspect that errors would be differential by RTM use status. We included only two measures of quality of care. A more thorough evaluation of quality-of-care measures (which was outside the scope of this study) may have resulted in a more clear and consistent association with RTM facility volume. We assumed that those not enrolled were not offered the opportunity to enroll, but some patients refuse RTM, though this information is not tracked, and refusals are relatively uncommon. Lastly, we only evaluated the reach and adoption components of the RE-AIM framework. Effectiveness was evaluated separately [25]. It would be useful for future studies to assess implementation and maintenance. Important strengths of this study include its national scope and ascertainment of information beyond medical conditions, including characteristics of the patient's social environment (e.g., Area Deprivation Index) and healthcare access and utilization, to provide a more comprehensive picture. Furthermore, our multivariable model allowed us to evaluate the independent associations between factors and RTM enrollment.

In conclusion, this study provides important information on the adoption and reach of RTM in the initial years of national implementation in VHA, including characteristics of facilities with higher and lower use, and characteristics of enrolled patients relative to those who are targeted. Our findings indicate that adoption of RTM was not equitable as this innovation was primarily employed by higher-complexity facilities with patients who had better geographic access and who lived in areas with less deprivation. Furthermore, within facilities employing RTM, there were indications of inequitable distribution of RTM. It will be valuable to continue to identify barriers to RTM and potential solutions to ensure that this technology is equitably disseminated. Equity will require matching of RTM need and use across facilities and continued surveillance for disparities.

## Supporting information

**S1 Table. International Classification of Diseases, 10<sup>th</sup> edition (ICD-10) codes for ulceration.**
(PDF)

## Acknowledgments

The views expressed in this article are those of the authors and do not necessarily reflect the position or policy of the Department of Veterans Affairs or the United States government. A portion of this paper was presented as a poster at the International Symposium of the Diabetic Foot (the Hague Netherlands) in May 2023.

## Author Contributions

**Conceptualization:** Alyson J. Littman, Andrew K. Timmons, Suzanne Shirley, Ernest Moy.

**Data curation:** Andrew K. Timmons, Kenneth T. Jones.

**Formal analysis:** Andrew K. Timmons, Kenneth T. Jones.

**Funding acquisition:** Alyson J. Littman.

**Investigation:** Alyson J. Littman, Suzanne Shirley, Ernest Moy.

**Methodology:** Alyson J. Littman, Andrew K. Timmons, Kenneth T. Jones, Suzanne Shirley, Jeffrey Robbins, Ernest Moy.

**Project administration:** Alyson J. Littman.

**Resources:** Alyson J. Littman, Ernest Moy.

**Software:** Andrew K. Timmons, Kenneth T. Jones.

**Supervision:** Alyson J. Littman, Ernest Moy.

**Validation:** Alyson J. Littman.

**Writing – original draft:** Alyson J. Littman.

**Writing – review & editing:** Alyson J. Littman, Andrew K. Timmons, Kenneth T. Jones, Suzanne Shirley, Jeffrey Robbins, Ernest Moy.

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
