## [Decision Letter · Decision Letter 0]

5 Nov 2023

PONE-D-23-29049Assessing equity in the uptake of remote foot temperature monitoring in a large integrated US healthcare systemPLOS ONE

Dear Dr. Littman,

Thank you for submitting your manuscript to PLOS ONE. After careful consideration, we feel that it has merit but does not fully meet PLOS ONE’s publication criteria as it currently stands. Therefore, we invite you to submit a revised version of the manuscript that addresses the points raised during the review process.

We look forward to receiving your revised manuscript.

Kind regards,

Yoshihisa Tsuji

Academic Editor

PLOS ONE

Journal Requirements:

 "This work was supported in part by the Department of Veterans Affairs (VA) Health Services Research and Development and the VA Office of Health Equity (CIN 13-402)."

4. We note that Figure 2 in your submission contain map images which may be copyrighted. All PLOS content is published under the Creative Commons Attribution License (CC BY 4.0), which means that the manuscript, images, and Supporting Information files will be freely available online, and any third party is permitted to access, download, copy, distribute, and use these materials in any way, even commercially, with proper attribution. For these reasons, we cannot publish previously copyrighted maps or satellite images created using proprietary data, such as Google software (Google Maps, Street View, and Earth). For more information, see our copyright guidelines: http://journals.plos.org/plosone/s/licenses-and-copyright.

Reviewers' comments:

Reviewer's Responses to Questions

**Comments to the Author**

1. Is the manuscript technically sound, and do the data support the conclusions?

Reviewer #1: Partly

Reviewer #2: Partly

2. Has the statistical analysis been performed appropriately and rigorously? 

Reviewer #1: I Don't Know

Reviewer #2: Yes

3. Have the authors made all data underlying the findings in their manuscript fully available?

Reviewer #1: Yes

Reviewer #2: Yes

4. Is the manuscript presented in an intelligible fashion and written in standard English?

Reviewer #1: Yes

Reviewer #2: Yes

5. Review Comments to the Author

Reviewer #1: This manuscript addresses a translational study of RTM management for diabetic foot ulcers.

The authors surveyed a national scope utilizing VHA's electronic health records.

The items examined included not only medical condition but also patient’s social environment and healthcare access.

The study is significant in considering the application of RTM to the general practice.

There are several points of concern with the methodology, and I will comment on them.

1.

In this manuscript, Adoption of RTM is reviewed on an individual patient basis. However, it is Reach that originally evaluates utilization at the individual level.

In the evaluation of adoption, isn't the utilization rate evaluated at the facility level?

2.

I would like to confirm the statistical analysis in the validation of Table 2.

The Method states that the test between each RTM use category was tested using the chi-square test.

There is a concern about multiple testing because of the comparison among the four groups.

Please specify clearly which groups were compared.

Reading the results, it appears that the chi-square test was only performed between the No patients facility and the high volume facility.

3.

The Manuscript presents patient/podiatrist ratios as facility characteristics, but no data on patient volume is published.

Since this facility type classification is determined by the absolute number of RTM-registered patients, facilities that simply have a higher total number of patients tend to be High volume.

Line 139 describes the reason for using the absolute number of patients rather than the percentage of patients using RTMs, but I would like to see a clearer explanation added.

4.

It is noted that the COVID19 outbreak has had an impact on RTM registered patients.

This study was conducted both before and after the COVID19 epidemic, which may have affected facility and patient characteristics.

I think COVID19 is a factor that may have affected the consistency of the results.

Please mention the impact of COVID19 on the results.

5.

There are some considerations regarding Limitation.

Since this is a database study, it is desirable to specify that data on individual characteristics (e.g., HbA1c % data, diabetes-related treatment, etc.) are not available.

Reviewer #2: The retrospective, observational study from the US health care system, the Veterans Health Administration (VHA) by Dr. Littman AJ et al investigates the equity in the uptake of remote foot temperature monitoring (RTM) to prevent amputation for DM with foot ulceration. The use of RTM increased substantially but the growth was concentrated in a small number of higher-resourced facilities and the less use of RTM in patients with Black and liver farther from specialty care.

Although the reviewer agrees with the importance of monitoring RTM as a diagnostic tool for preventing amputation and the data could provide a clinically significant message, the reviewer found some concerns in the study.

The reviewer’s specific comments are as follows:

＜Major concerns＞

1. Through discussion of the results, the analysis and interpretation of the heterogeneity of RTM use is ambiguous, and difficult to understand the message of the study. Why not present and discuss the results separately for factors on the facility side, factors on the prescribing doctor (including patient selection), and factors on the patient's side who accepts the prescription?　

2. Although the authors compared characteristics of patients and controls to analyze heterogeneity in RTM use (Table 3), selection criteria may differ between institutions; The reviewer believes that excluding only facility patients without RTM use is not appropriate for the selection of comparison group. Since facilities are already categorized by patient volume in Table 2, it would be better to match them based on gender, age, Low, Medium, and High facilities, and the timing of RTM enrolment.

3. The authors recently reported prognosis using RTM in VAH, which the reviewers consider to be the same population as this manuscript (Diabetes Care 2023;46:1464-1468). In contrast to previous reports, RTM did not reduce the incidence of lower extremity amputation or all-cause hospitalization. According to this study, RTM appears to be well-enrolled in high-risk patients, but is it possible that there was a problem with patient selection and that patient prognosis did not improve as a result? Please discuss.

＜Minor concerns＞

１．This journal has no limit to the number of tables. What is the reason to make ST2 as Supplemental material? If there is no specific reason, use it as a regular table and carefully analyze and interpret the data.

2. Please add information about the adoption of RTM in the Introduction so that readers in countries/facilities that have not used RTM can understand. The reviewer thinks it would be good to have information such as whether there are qualifications and standards for prescribing doctors when implementing RTM, and the extent of the financial burden on patients (prescription fees, diagnostic fees, internet environment maintenance costs, etc.).

6. PLOS authors have the option to publish the peer review history of their article (what does this mean?). If published, this will include your full peer review and any attached files.

Reviewer #1: No

Reviewer #2: No

---

## [Author Response · Author response to Decision Letter 0]

24 Feb 2024

We thank the editor and reviewers for their thoughtful review. Below, we respond to each critique.

Sincerely,

Alyson Littman, on behalf of the authors

Reviewer #1: This manuscript addresses a translational study of RTM management for diabetic foot ulcers.

The authors surveyed a national scope utilizing VHA's electronic health records.

The items examined included not only medical condition but also patient’s social environment and healthcare access.

The study is significant in considering the application of RTM to the general practice.

There are several points of concern with the methodology, and I will comment on them.

** Response: We appreciate the reviewer noting some of the strengths of our study.

1. In this manuscript, Adoption of RTM is reviewed on an individual patient basis. However, it is Reach that originally evaluates utilization at the individual level.

In the evaluation of adoption, isn't the utilization rate evaluated at the facility level?

** Response: As the reviewer expected, the utilization rate was evaluated at the facility level. In lines 120-125, we define adoption as “the number and proportion of settings and staff members that agree to initiate program or policy change and how representative they are of the intended audience. We evaluated adoption by first describing the geographic distribution of RTM throughout VHA over time and then assessing difference in characteristics of patients who received care at facilities with higher vs. lower (or no) use of RTM. These adoption analyses aimed to assess whether there were differences in facility-level characteristics based on RTM use. In the methods (lines 156-160) we describe the categories we created, “To examine adoption, we classified facilities based on the proportion of eligible patients enrolled in RTM during the study time period into one of four categories: no RTM use, low RTM use (<2% of eligible patients enrolled), moderate RTM use (2-<10% of eligible patients enrolled), and high RTM use (>10% of eligible patients enrolled).”

2. I would like to confirm the statistical analysis in the validation of Table 2.

The Method states that the test between each RTM use category was tested using the chi-square test.

There is a concern about multiple testing because of the comparison among the four groups. Please specify clearly which groups were compared.

Reading the results, it appears that the chi-square test was only performed between the No patients facility and the high volume facility.

** Response: We apologize for the confusion. The chi-square test assessed for a difference across all categories. In the revised manuscript, we explain on lines 170-176, “To assess whether there was equitable use of RTM across facilities, we examined distributions of patient demographic, geographic, and facility characteristics across facility RTM use categories (e.g., no RTM use, and low, moderate, and high RTM use) and calculated chi-square statistics to test the statistical significance of differences. Because statistical tests with so many comparisons and a large sample may not be informative in terms of meaningful differences, we highlighted meaningful differences between categories (see Results).” In the results, we also now state in lines 248-250: “Here, we highlight larger absolute differences across categories (>5 percentage points between high and no patient categories).” 

3. The Manuscript presents patient/podiatrist ratios as facility characteristics, but no data on patient volume is published.

Since this facility type classification is determined by the absolute number of RTM-registered patients, facilities that simply have a higher total number of patients tend to be High volume.

Line 139 describes the reason for using the absolute number of patients rather than the percentage of patients using RTMs, but I would like to see a clearer explanation added.

** Response: In light of the reviewer’s critique, we have revised our approach. We now classify facilities based on the proportion of eligible patients enrolled. The manuscript now states, “… we classified facilities based on the proportion of eligible patients enrolled in RTM during the study time period into one of four categories: no RTM use, low RTM use (<2% of eligible patients enrolled), moderate RTM use (2-<10% of eligible patients enrolled), and high RTM use (>10% of eligible patients enrolled).” This resulted in some facilities being reclassified, but results were similar. In cases where there were changes (e.g., race), we revised the abstract, results, and discussion.

4. It is noted that the COVID19 outbreak has had an impact on RTM registered patients.

This study was conducted both before and after the COVID19 epidemic, which may have affected facility and patient characteristics.

I think COVID19 is a factor that may have affected the consistency of the results.

Please mention the impact of COVID19 on the results.

** Response: We appreciate this suggestion. In the revised manuscript, we note that there was a steep increase in the number of patients enrolled per month concurrent with the start of the COVID-19 pandemic (lines 216-219). We also added to the discussion (lines 314-320) about why enrollment may have increased during the pandemic.

5. There are some considerations regarding Limitation.

Since this is a database study, it is desirable to specify that data on individual characteristics (e.g., HbA1c % data, diabetes-related treatment, etc.) are not available.

** Response: The reviewer makes a good point. We now note that because this was a study that relied on medical record data, data on factors like diabetes-related treatment was not available or feasible to extract. Please note that we were able to extract hemoglobin A1c levels; this was included in Table 3.

Reviewer #2: The retrospective, observational study from the US health care system, the Veterans Health Administration (VHA) by Dr. Littman AJ et al investigates the equity in the uptake of remote foot temperature monitoring (RTM) to prevent amputation for DM with foot ulceration. The use of RTM increased substantially but the growth was concentrated in a small number of higher-resourced facilities and the less use of RTM in patients with Black and liver farther from specialty care.

Although the reviewer agrees with the importance of monitoring RTM as a diagnostic tool for preventing amputation and the data could provide a clinically significant message, the reviewer found some concerns in the study.

The reviewer’s specific comments are as follows:

＜Major concerns＞

1. Through discussion of the results, the analysis and interpretation of the heterogeneity of RTM use is ambiguous, and difficult to understand the message of the study. Why not present and discuss the results separately for factors on the facility side, factors on the prescribing doctor (including patient selection), and factors on the patient's side who accepts the prescription?　

** Response: Thank you for this feedback. In the revised manuscript, we tried to make clearer that the analyses presented in Table 2 attempted to understand factors associated with RTM use at a facility level. Though most of the factors assessed in these analyses were facility- (e.g., VA district, facility complexity, patients per podiatrist, AK/BK ratio, % PAVE high risk, Percent of patients with documented monofilament exam) or area- (e.g., area deprivation index, broadband access) level measures, we also thought it was valuable to present information on the characteristics of patients at those facilities (e.g., race, ethnicity, drive time). Unfortunately, we did not have information to determine which patients refused RTM enrollment (i.e., offered enrollment but declined). Table 3 focuses on patient characteristics that may inform us about patient selection, such as age, sex, enrollment priority, foot conditions, and healthcare utilization. Six variables (race, ethnicity, drive distance to primary care and drive time to primary and specialty care, and poor broadband access) were included in both analyses. We felt that these factors were important to consider in terms of facilities offering RTM and within the facilities offering RTM, the specific patients enrolled.

2. Although the authors compared characteristics of patients and controls to analyze heterogeneity in RTM use (Table 3), selection criteria may differ between institutions; The reviewer believes that excluding only facility patients without RTM use is not appropriate for the selection of comparison group. Since facilities are already categorized by patient volume in Table 2, it would be better to match them based on gender, age, Low, Medium, and High facilities, and the timing of RTM enrolment.

** Response: We appreciate the reviewer’s suggestion and evaluated whether associations between the factors assessed, and RTM enrollment differed based on the RTM facility classifications used in Table 2. Generally, results were consistent across the strata, indicating that matching or adjusting for facility classifications would not meaningfully change the results or interpretation of findings. In the revised manuscript, we now discuss this sensitivity analysis in the methods (lines 188-192).

3. The authors recently reported prognosis using RTM in VAH, which the reviewers consider to be the same population as this manuscript (Diabetes Care 2023;46:1464-1468). In contrast to previous reports, RTM did not reduce the incidence of lower extremity amputation or all-cause hospitalization. According to this study, RTM appears to be well-enrolled in high-risk patients, but is it possible that there was a problem with patient selection and that patient prognosis did not improve as a result? Please discuss.

** Response: We have added a paragraph to the introduction (lines 98-110) to address the reviewer’s concern. Briefly, we discuss the conflicting literature and possible reasons for the discrepancies across studies.

＜Minor concerns＞

１．This journal has no limit to the number of tables. What is the reason to make ST2 as Supplemental material? If there is no specific reason, use it as a regular table and carefully analyze and interpret the data.

** Response: We now include previously named Supplemental Table 2 as Table 4.

2. Please add information about the adoption of RTM in the Introduction so that readers in countries/facilities that have not used RTM can understand. The reviewer thinks it would be good to have information such as whether there are qualifications and standards for prescribing doctors when implementing RTM, and the extent of the financial burden on patients (prescription fees, diagnostic fees, internet environment maintenance costs, etc.).

** Response: We appreciate this suggestion and have added a paragraph to the introduction on the qualifications and standards for prescribing doctors when implementing RTM, and the extent of the financial burden on patients.

---

## [Editor Report · Decision Letter 1]

14 Mar 2024

Assessing equity in the uptake of remote foot temperature monitoring in a large integrated US healthcare system

PONE-D-23-29049R1

Dear Dr. Littman,

We’re pleased to inform you that your manuscript has been judged scientifically suitable for publication and will be formally accepted for publication once it meets all outstanding technical requirements.

Kind regards,

Yoshihisa Tsuji

Academic Editor

PLOS ONE
---

## [Editor Report · Acceptance letter]

22 Mar 2024

PONE-D-23-29049R1 

PLOS ONE

Dear Dr. Littman, 

I'm pleased to inform you that your manuscript has been deemed suitable for publication in PLOS ONE. Congratulations! Your manuscript is now being handed over to our production team.

Kind regards, 

on behalf of

Professor Yoshihisa Tsuji 

Academic Editor

PLOS ONE